# The Impact of Single-Event Radiation on Latch-Up Effect in High-Temperature CMOS Devices and Its Mechanism

**DOI:** 10.3390/mi16070783

**Published:** 2025-06-30

**Authors:** Bin Wang, Jianguo Cui, Ling Lv, Longsheng Wu

**Affiliations:** State Key Laboratory of Wide Bandgap Semiconductor Technology, Faculty of Integrated Circuit, XiDian University, Xi’an 710071, China; wbin@xidian.edu.cn (B.W.);

**Keywords:** complementary metal-oxide-semiconductor (CMOS), single-event latch-up (SEL), high temperature, linear energy transfer (LET)

## Abstract

This paper investigates the latch-up effect in CMOS devices based on a 28 nm CMOS process within the temperature range of 200 K to 450 K using *Sentaurus* Technology Computer-Aided Design (TCAD) simulation, with a particular focus on the single-event latch-up (SEL) effect in the high-temperature range of 300 K to 450 K. The physical mechanism underlying the triggering of SEL in CMOS devices at high temperatures is revealed. The results show that when the linear energy transfer (LET) value is 75 MeV cm^2^/mg, the CMOS devices do not exhibit SEL effects at 300 K and 350 K. However, when the temperature rises to 400 K, a significant latch-up effect occurs, which becomes more pronounced with increasing temperature. Additionally, at a supply voltage of 1.2 V and a temperature of 450 K, the LET threshold for triggering SEL in CMOS devices decreases by 91.4% compared to 75 MeV cm^2^/mg at 300 K, dropping to 6 MeV cm^2^/mg. As the temperature increases, the latch-up trigger current of the CMOS devices decreases from 1.18 × 10^−4^ A/μm at 300 K to 4.65 × 10^−5^ A/μm at 450 K, and the hold voltage decreases from 1.48 V at 300 K to 1.07 V at 450 K.

## 1. Introduction

In space environments such as near-Earth orbits and the lunar surface, CMOS integrated circuits are exposed to significant temperature variations. The surface facing the sun can reach temperatures above 150 °C (423 K), while the shaded side can drop to below −100 °C (173 K) [1]. However, the latch-up effect in CMOS devices is highly sensitive to temperature variations. The latch-up effect in CMOS devices at low temperatures has been extensively studied, yielding a series of meaningful results [2,3,4,5,6]. Concurrently, the single-event latch-up (SEL) effect under elevated-temperature conditions is attracting growing research interest.

In fact, as integrated circuit technology enters the deep sub-micron era, the high-temperature characteristics of MOSFET devices have changed significantly. The threshold voltage decreases, the off-state current deteriorates, the carrier mobility reduces, and the power consumption of the circuit increases, severely affecting the performance of CMOS integrated circuits and potentially causing functional failures [7]. The temperature dependence of CMOS latch-up has been widely studied. J.G. Dooley et al. investigated the temperature dependence of latch-up hold current, trigger current, and bulk resistance in CMOS devices, finding that the hold and trigger currents decrease with increasing temperature, while the CMOS bulk resistance and parasitic transistor current gain increase [8]. E. Sangiorgi et al. studied the temperature dependence of latch-up hold voltage in very large-scale integration (VLSI) CMOS technology over the range of 77 K to 400 K, observing a significant increase in CMOS parasitic bulk resistance with rising temperature, leading to an increase in hold voltage [9]. C.C. Yao et al. investigated the temperature dependence of latch-up in asymmetric lightly doped drain (LDD) CMOS, proposing that the hold current decreases with increasing temperature and the trigger voltage slightly decreases as temperature rises [10]. They also provided a relationship between the hold current, parasitic transistor current gain, and parasitic bulk resistance. Liang et al. studied the temperature characteristics of electrostatic discharge (ESD) protection devices in the high-temperature range of 300 K to 500 K, finding that the hold voltage and hold current decrease with increasing temperature, increasing the risk of latch-up in ESD protection devices at high temperatures [11]. These studies reveal that CMOS latch-up effects exhibit pronounced temperature dependence.

However, the above research exclusively addresses conventional latch-up phenomena without addressing single-particle radiation impacts. Actually, the SEL effects induced by the radiation become significantly more pronounced under elevated temperatures. A. H. Johnston et al. investigated the impact of elevated temperatures on SEL effects. The study revealed that latch-up thresholds are strongly influenced by contact geometry, with temperature dependence consistent with models based on triggering vertical parasitic transistors. At 125 °C, the linear energy transfer (LET) threshold decreased by a factor of approximately 2.5 relative to room temperature [12]. H. Iwata et al. investigated the temperature dependence of SEL phenomena in CMOS devices across the 77–450 K temperature range. The study demonstrated that CMOS immunity to SEL progressively diminishes as temperature increases from 120 K to 450 K [13]. S. Guagliardo et al. investigated the impact of temperature variations on SEL susceptibility in CMOS devices. The investigation considered three geometry and design-dependent parameters: doping profiles, anode-to-cathode (A–C) spacing, and placement of substrate/well contacts. The study revealed that temperature effects on SEL susceptibility become more pronounced in devices exhibiting lower inherent SEL sensitivity [14]. E. V. Mrozovskaya et al. experimentally investigated the relationship between SEL cross-section and LET for various CMOS circuits across different temperatures. The experimental results demonstrated that the SEL cross-section increases with elevated temperatures [15]. However, these studies have primarily focused on the impact of parameters such as CMOS device process manufacturing, doping profiles, and anode-to-cathode spacing on SEL effects, while the physical mechanisms triggering SEL in CMOS devices at elevated temperatures have not been sufficiently explored. Furthermore, with the continued scaling of CMOS device feature sizes, the characteristics of 28 nm process CMOS devices under single-particle irradiation differ significantly from those with larger feature sizes [16].

Therefore, this paper investigates the high-temperature characteristics of single-event latch-up in CMOS devices based on the 28 nm process over the temperature range of 300 K to 450 K, using Sentaurus TCAD simulation to reveal the physical mechanism underlying the triggering of SEL in CMOS devices at high temperatures based on the generation and recovery processes of excess carriers. The study also explores the sensitivity factors of SEL effects.

## 2. Device Parameters and Model Establishment

Figure 1 shows the structure of the CMOS device, with parameters designed based on 28 nm CMOS technology. The P-type substrate doping concentration is 1 × 10^16^ cm^−3^, the N-Well doping concentration is 1 × 10^18^ cm^−3^, and the source and drain doping concentrations of both NMOS and PMOS are 1 × 10^20^ cm^−3^. Detailed parameters are listed in Table 1.

Figure 2 illustrates the equivalent circuit diagram of the CMOS single-event latch-up effect [17,18]. R_n_ and R_p_ represent the parasitic bulk resistances of the N-Well and P-substrate, respectively, while R_n1_ and R_p1_ are the internal extended resistances of the N-Well and P-substrate, which act as shunts for the emitter junctions. T1 and T2 are the parasitic transistors within the well and substrate. To study the SEL effect in high-temperature environments, a heavy ion model was added to the Sentaurus TCAD software. A single heavy ion is incident on the sensitive region of the NMOS drain when the CMOS device input is at low voltage [19]. The ion incidence direction is along the positive Y-axis, with a radiation time of 1 × 10^−11^ s. The A-A’ cutline indicates the direction along the heavy ion incidence. Additionally, the following models were included: Fermi model, bandgap narrowing model, SRH (Shockley–Read–Hall recombination) and Auger recombination models, high-field saturation effect model, doping-dependent scattering, interface scattering, and carrier–carrier scattering [13,14].

## 3. Simulation Results and Analysis

Figure 3 describes the impact of single-event radiation on the latch-up effect in CMOS devices at different temperatures ranging from 300 K to 450 K. The LET value of the heavy ion is 75 MeV cm^2^/mg, and the supply voltage is 1.2 V. At 0.01 ns, a heavy ion strikes the drain side of the NMOS, and the internal current of the CMOS device is monitored after the ion strike. The transient scan duration is 6 ns. Before the heavy ion incidence (t < 0.01 ns), the internal current of the CMOS device remains at the μA level. After the ion incidence (t = 0.01 ns), a transient current pulse appears due to the generation of a large number of non-equilibrium carriers, as shown in Figure 3. At 300 K, the peak pulse current is 2.0 mA, and it increases to 2.1 mA at 350 K, 2.2 mA at 400 K, and 2.4 mA at 450 K with rising temperature. Additionally, the pulse current at 300 K and 350 K returns to the μA level before ion incidence (t < 0.01 ns) after 2 ns and 3.5 ns, respectively, indicating that the latch-up effect does not occur. However, when the temperature rises to 400 K and 450 K, the pulse current remains at the 10^−4^ level after 6 ns, significantly higher than the current before ion incidence, resulting in a tail current (latch-up current), indicating that the latch-up effect is triggered. Moreover, the latch-up current at 450 K is 9.5 × 10^−4^ A/μm, higher than the latch-up current at 400 K (5.6 × 10^−4^ A/μm), demonstrating that the SEL effect becomes more pronounced with increasing temperature. The reasons why the SEL effect is more severe at high temperatures are mainly twofold. First, the concentration of non-equilibrium carriers generated by single-particle irradiation increases significantly at high temperatures. Second, the latch-up threshold of CMOS devices decreases with the rise in temperature. The following analysis will focus on these two aspects.

Figure 4 depicts the electron distribution within the device at different temperatures ranging from 300 K to 450 K before ion incidence (t = 0.005 ns). At t = 0.005 ns, carriers drift under the influence of the supply voltage. Due to the decrease in ionization energy with increasing temperature, the electron density in the P-Sub region increases from 10^4^ cm^−3^ at 300 K to 10^11^ cm^−3^ at 450 K, as illustrated in Figure 4, increasing the current before ion incidence (t = 0.005 ns). Figure 5 shows the electron distribution within the device at different temperatures ranging from 300 K to 450 K after ion incidence (t = 0.01 ns). At t = 0.01 ns, a large number of excess carriers are generated within the ion incidence track, and the concentration of excess carriers increases significantly with rising temperature, from 4.0 × 10^18^ cm^−3^ at 300 K to 6 × 10^18^ cm^−3^ at 450 K, as shown in Figure 5. The large number of excess carriers generated by irradiation creates a transient current pulse, as shown in Figure 3. Since the pulse current is positively correlated with the concentration of excess carriers, the peak pulse current increases significantly from 2.1 mA to 2.4 mA with rising temperature, as shown in Figure 3.

Figure 5b describes the variation in excess carrier concentration at point C concerning time. The variation in current within the device, as shown in Figure 3, is consistent with the variation in excess carriers generated by irradiation. At 300 K, the excess carriers return to equilibrium after 2 ns. Therefore, the pulse current at 300 K decreases from a peak of 2.1 mA to the μA level before irradiation after 2 ns. As the recovery time of excess carriers increases with their concentration, the recovery time at 350 K increases to 3.5 ns. When the temperature rises to 400 K and 450 K, the carrier concentration remains at the 10^18^ level after irradiation, significantly higher than the carrier concentration before irradiation. The high concentration of excess carriers triggers the parasitic thyristor in the CMOS structure, causing the latch-up effect and forming a latch-up current. Moreover, the latch-up current at 450 K is higher than that at 400 K due to the higher carrier concentration after irradiation, as shown in Figure 3.

Figure 6 shows the current density distribution within the CMOS device at different temperatures ranging from 300 K to 450 K after ion incidence (t = 6 ns). At 300 K and 350 K, the device returns to equilibrium, and the current density after irradiation is 10^1^ A × cm^−2^, indicating that the latch-up effect does not occur. However, when the temperature rises to 400 K, the current density along the B-B′ cutline increases significantly, with the highest current density near the ion incidence point, forming a latch-up current. Moreover, the current density at 450 K increases significantly with rising temperature, leading to a more severe latch-up effect, as shown in Figure 6. This proves that the SEL effect in CMOS devices is positively correlated with temperature.

Figure 7 shows the equivalent circuit diagram of the lumped element model for CMOS latch-up [20]. The trigger voltage, trigger current, hold voltage, and hold current can be calculated from the equivalent circuit as follows:(1)$$V_{trig}=V_{DD}+V_{bi1}+V_{bi2}\times \frac{R_{ew}}{R_{p}\times \partial_{PNP}}$$(2)$$I_{trig}=\frac{V_{bi2}}{R_{p}\times \partial_{PNP}}\;\;$$(3)$$I_{h}=\frac{R_{es}\times V_{DD}+R_{es}\times V_{bi1}+R_{n}\times V_{bi2}}{R_{es}\times \partial_{PNP}\left(R_{p}+R_{p1}\right)+\partial_{PNP}\left(R_{p}+R_{n}\right)}\;$$(4)$$V_{h}=I_{h}\times \left(R_{ew}+R_{p}\times \partial_{PNP}+R_{p1}\times \partial_{PNP}\right)+V_{CE1}\approx V_{bi2}\times \left(1+\frac{R_{P1}}{R_{p}}\right)+V_{CE1}$$
where *β* is the common-base current gain of the *PNP* transistor; *R_P_*, *R_n_*, *R_es_*, and *R_ew_* are the parasitic bulk resistances of the CMOS device; *V_CE_*_1_ is the saturation voltage of the *PNP* transistor; and *V_bi_*_1_ and *V_bi_*_2_ are the built-in potentials of the emitter junctions of the parasitic transistors *T1* and *T2*. Since the parasitic bulk resistance and transistor current gain have positive temperature coefficients, while the built-in potential has a negative temperature coefficient, the latch-up trigger voltage (*V_trig_*), trigger current (*I_trig_*), hold current (*I_h_*), and hold voltage (*V_h_*) all decrease with increasing temperature.

Figure 8 illustrates the I-V characteristic curve of the CMOS latch-up effect over the temperature range of 200 K to 450 K. The point where the device transitions from the high-resistance off-state to the negative resistance region is the trigger point, corresponding to the trigger current (*I_trig_*) and trigger voltage (*V_trig_*). The point where the device transitions from the negative resistance region to the latch-up region is the hold point, corresponding to the hold voltage (*V_h_*) and hold current (*I_h_*). The trigger current and hold voltage determine whether the latch-up effect occurs. As the temperature increases from 200 K to 450 K, the latch-up trigger current, trigger voltage, hold voltage, and hold current all decrease, as shown in Figure 8.

Table 2 presents the simulation results of *I_trig_*, *V_trig_*, *V_h_*, and *I_h_* over the temperature range of 200 K to 450 K. As the temperature increases, *I*_trig_ decreases from 1.18 × 10^−4^ A at 300 K to 4.65 × 10^−5^ A at 450 K, while *V_h_* decreases from 1.48 V at 300 K to 1.07 V at 450 K. The reduction in latch-up threshold increases the risk of latch-up occurrence in CMOS devices at high temperatures. The supply voltage of 1.2 V is lower than the hold voltage (*V_h_*) at 300 K and 350 K, failing to reach the latch-up threshold at these temperatures. Consequently, the pulse current induced by irradiation decays to the μA level after 2 ns and 3.5 ns, respectively, without triggering latch-up. However, as the temperature rises to 400 K and 450 K, the hold voltage (*V_h_*) drops to 1.18 V and 1.07 V, respectively. The supply voltage of 1.2 V then reaches the latch-up threshold, thereby triggering the latch-up effect, which becomes more pronounced with increasing temperature, as shown in Figure 3.

Following the heavy ion strike, the carrier generation function of the device follows the Gaussian distribution with a 10 nm radius and characteristic time of (*T_C_* = 10 ps) [21]. LET represents the charge deposition per unit length along the radiation track, with units of MeV·cm^2^/mg. The impact of heavy ions is carried out through another parameter known as ion generation rate (*G_rion_*). It is calculated by mathematical expressions.(5)$$G_{rion}=G_{rLET}\times R_{iON}\times T_{iON}\;$$

Here, *G_rLET_* is the LET density, and *R_iON_* and *T_iON_* are functions for spatial and temporal distribution of the generation rate, respectively:(6)$$G_{rLET}=\frac{1}{\pi r_{0}^{2}}LET\_f(l)$$(7)$$R_{iON}\left(r,l\right)=exp\left(-{\left(\frac{r}{r_{0}}\right)}^{2}\right)$$(8)$$T_{iON}\left(t\right)=\frac{2\exp\left(-{\left(\frac{t-t_{0}}{\sqrt{2}W_{0}}\right)}^{2}\right)}{\sqrt{2}W_{0}\sqrt{\pi }\left(1+\mathrm{erf}\left(\frac{t_{0}}{\sqrt{2}W_{0}}\right)\right)}$$

*LET*_*f* is defined as the heavy ion model in picoCoulombs, *r* is the perpendicular radial distance from the ion track, *t* is time, and *r*_0_ and *W*_0_ are characteristic parameters of the Gaussian function [22,23,24].

Figure 9 describes the SEL effect in CMOS devices at 450 K for different LET values, namely 4 MeV cm^2^/mg, 5 MeV cm^2^/mg, 6 MeV cm^2^/mg, 6.1 MeV cm^2^/mg, and 6.3 MeV cm^2^/mg. When the LET is 4 MeV cm^2^/mg and 5 MeV cm^2^/mg, the pulse current returns to the μA level before ion incidence (t < 0.01 ns) after 1.6 ns and 2.1 ns, respectively, without triggering the latch-up effect. According to Equation (6), the concentration of excess carriers generated by heavy ions in silicon increases with the increasing LET value, as shown in Figure 10. Therefore, the transient current at 6 MeV cm^2^/mg is significantly higher than that at 4 MeV cm^2^/mg and 5 MeV cm^2^/mg, and the pulse current duration increases to 8 ns. However, the stable current value at 6 MeV cm^2^/mg is 12 μA/μm, higher than that at 4 MeV cm^2^/mg and 5 MeV cm^2^/mg. When the LET value further increases to 6.1 MeV cm^2^/mg and 6.3 MeV cm^2^/mg, the high concentration of excess carriers triggers the parasitic positive feedback loop in the CMOS structure, causing a sharp increase in current and triggering the latch-up effect, with the latch-up current increasing with the increasing LET value. Therefore, once the LET value of the incident heavy ion exceeds 6 MeV cm^2^/mg, the CMOS device will experience latch-up. Compared to the 75 MeV cm^2^/mg LET value at 300 K, which did not trigger latch-up, the latch-up LET threshold decreases by 91.4%.

## 4. Conclusions

This paper reveals the physical mechanism of the temperature dependence of latch-up effects in CMOS devices by deriving expressions for the latch-up trigger voltage, trigger current, hold voltage, and hold current. It also investigates the SEL effect in CMOS devices at high temperatures. The results show that LET value, supply voltage, and temperature are key factors in inducing the SEL phenomenon. With the continuous evolution of semiconductor technology and integrated circuit design, supply voltages are trending downward. For example, at the 28 nm process node, the supply voltage can be reduced to 1.2 V. Although low supply voltage helps protect against latch-up effects, rising temperatures, such as when a spacecraft moves from the shaded side to the sunlit side or when automotive-grade chips experience temperature increases during vehicle operation, can decrease the trigger current and hold voltage, thereby increasing latch-up sensitivity and severely affecting chip reliability. These in-depth analyses and discussions are of great significance for understanding and preventing SEL effects in CMOS devices at high temperatures.

## Figures and Tables

**Figure 1 micromachines-16-00783-f001:**
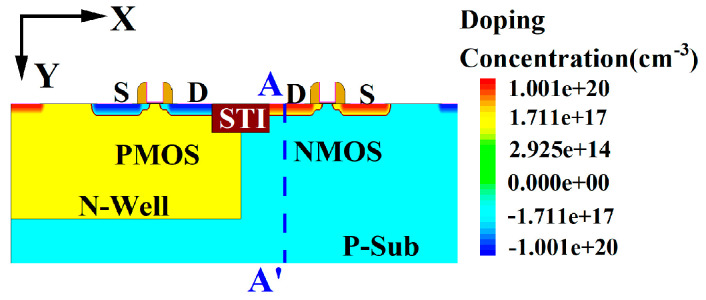
Structure of the CMOS device.

**Figure 2 micromachines-16-00783-f002:**
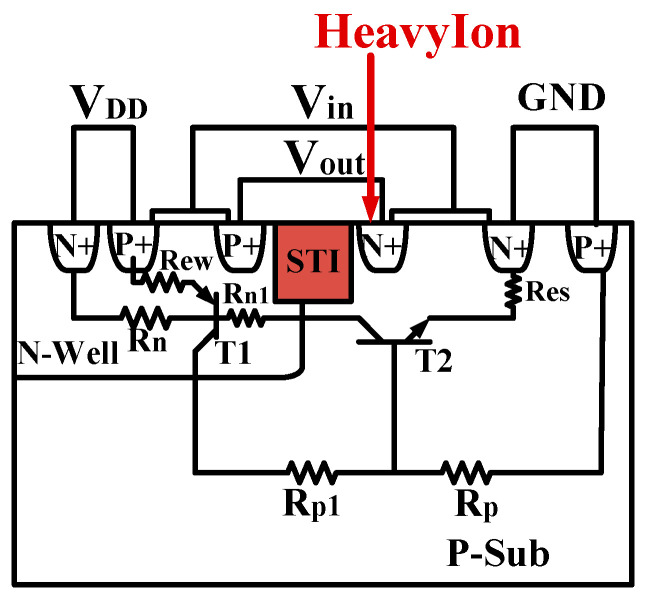
Equivalent circuit diagram of the CMOS single-event latch-up effect.

**Figure 3 micromachines-16-00783-f003:**
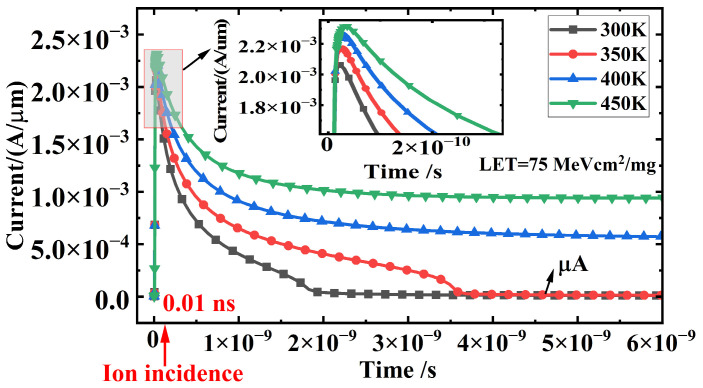
Current–time characteristic curves of single-event latch-up at 300–450 K.

**Figure 4 micromachines-16-00783-f004:**
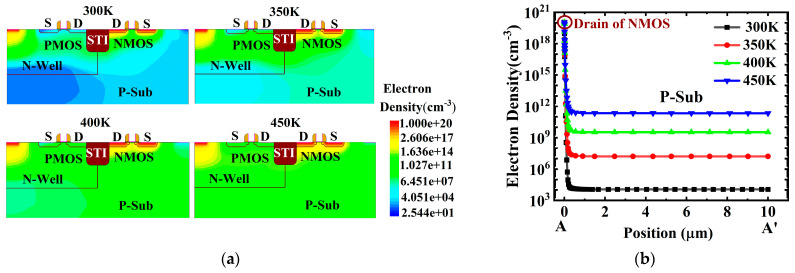
The electron distribution within the device at different temperatures ranging from 300 K to 450 K before ion incidence (t = 0.005 ns): (**a**) the electron density of the CMOS device; (**b**) the electron density of the CMOS device along the cutline A-A′.

**Figure 5 micromachines-16-00783-f005:**
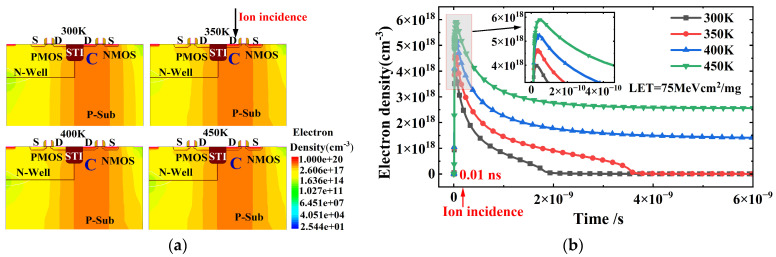
The electron distribution within the device at different temperatures ranging from 300 K to 450 K after ion incidence (t = 0.01 ns): (**a**) the electron density of the CMOS device; (**b**) the variation curves of electron density at point C for time.

**Figure 6 micromachines-16-00783-f006:**
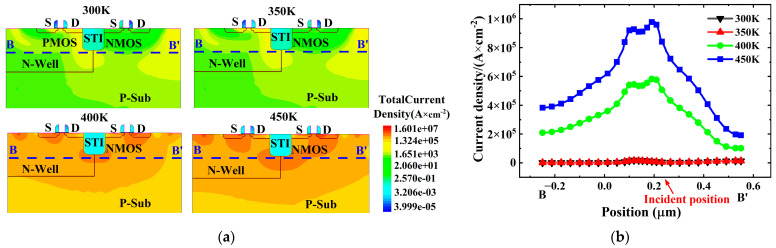
The total current density of CMOS device at different temperatures ranging from 300 K to 450 K after ion incidence (t = 6 ns): (**a**) the total current density of CMOS device; (**b**) the total current density of CMOS device along the cutline B-B′.

**Figure 7 micromachines-16-00783-f007:**
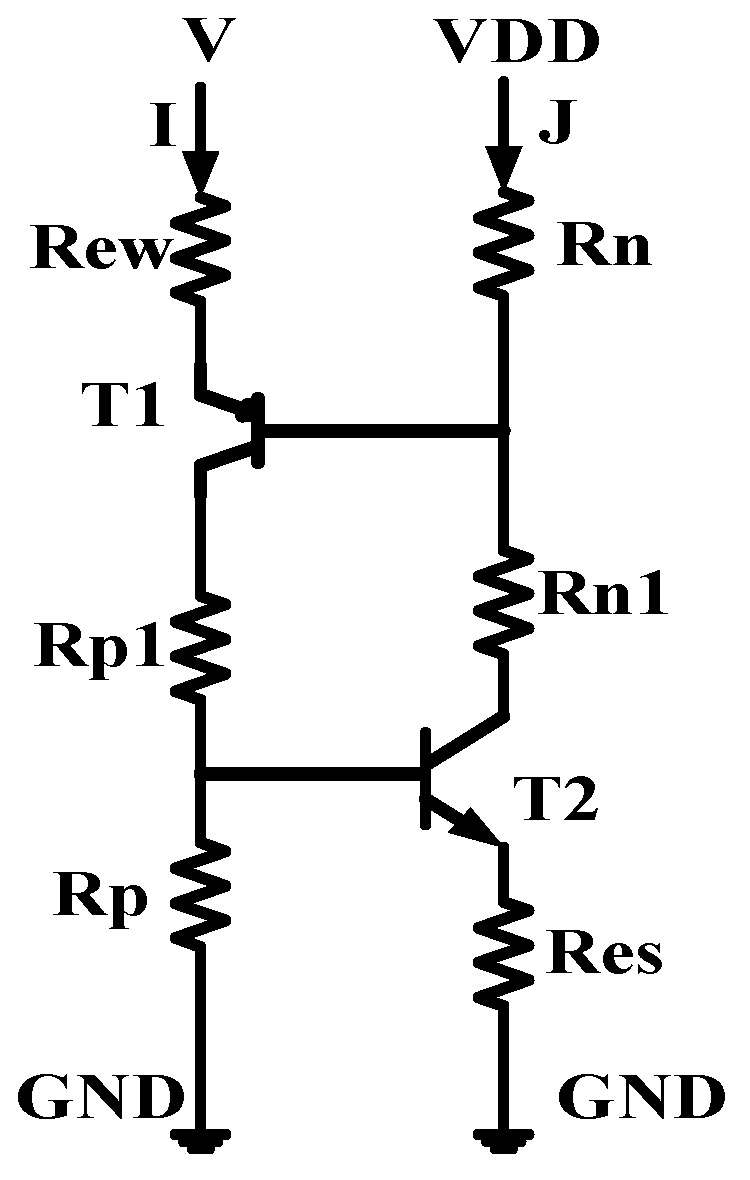
The equivalent circuit diagram of the lumped element model for CMOS latch-up.

**Figure 8 micromachines-16-00783-f008:**
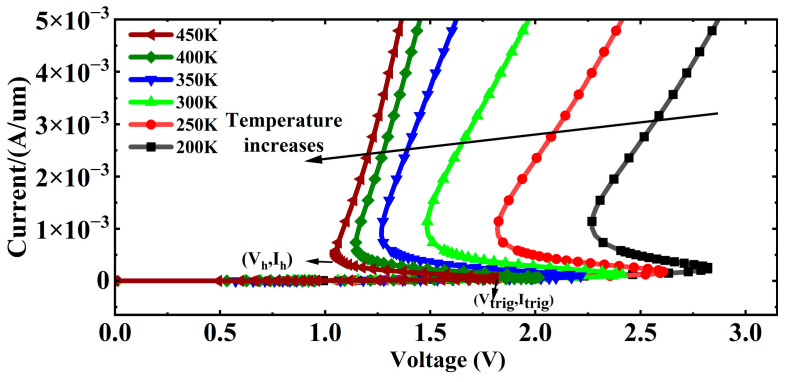
I–V characteristic curves of CMOS latch-up at different temperatures ranging from 200 K to 450 K.

**Figure 9 micromachines-16-00783-f009:**
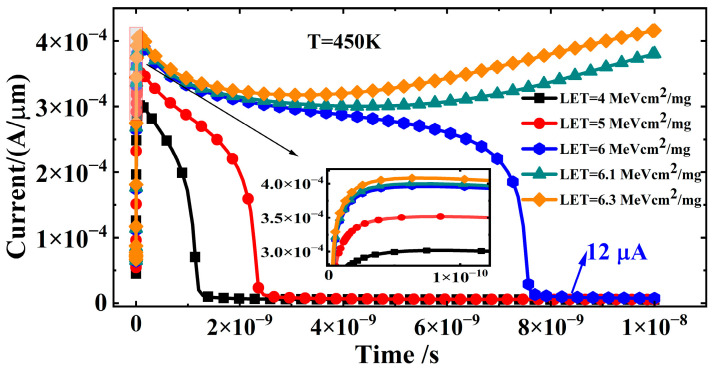
Current–time characteristic curves applying different LET values at 450 K.

**Figure 10 micromachines-16-00783-f010:**
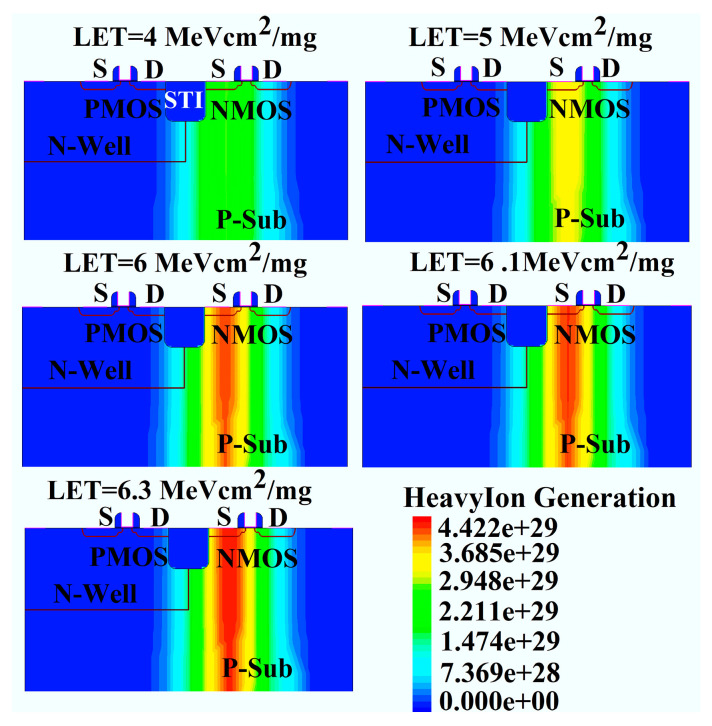
The distribution of the heavy ion generation applying different LET values at 450 K.

**Table 1 micromachines-16-00783-t001:** Parameters of the CMOS device.

Parameters	Values
Length of gate	28 nm
Source doping concentration	1 × 10^20^ cm^−3^
Drain doping concentration	1 × 10^20^ cm^−3^
LDD concentration	1 × 10^19^ cm^−3^
N-Well concentration	1 × 10^18^ cm^−3^
P-Sub concentration	1 × 10^16^ cm^−3^
Thickness of gate SiO_2_	1 nm
Thickness of gate HfO_2_	3 nm
Junction depth	20 nm
Thickness of silicon	10 um
Thickness of N-Well	200 nm

**Table 2 micromachines-16-00783-t002:** Parameters of characteristic curves at different temperatures.

Temperature (K)	*I_trig_* (A/μm)	*V_trig_* (V)	*V_h_* (V)	*I_h_* (A/μm)
450	4.65 × 10^−5^	1.81	data	4.02 × 10^−4^
400	6.85 × 10^−5^	2.02	1.18	5.75 × 10^−4^
350	9.14 × 10^−5^	2.21	1.27	7.28 × 10^−4^
300	1.18 × 10^−4^	2.41	1.48	9.15 × 10^−4^
250	1.72 × 10^−4^	2.60	1.82	1.03 × 10^−3^
200	2.47 × 10^−4^	2.82	2.26	1.21 × 10^−3^

## Data Availability

The original contributions presented in this study are included in the article. Further inquiries can be directed to the corresponding author(s).

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
