# Peer review of "The Impact of Single-Event Radiation on Latch-Up Effect in High-Temperature CMOS Devices and Its Mechanism"

_micromachines, 2025, doi:10.3390/mi16070783_

Round 1

Reviewer 1 Report

Comments and Suggestions for Authors

The paper investigates the SEL effect in CMOS devices fabricated with 28 nm process. Especially, the influence of high temperature of up to 450 K to the SEL sensitivity is researched. With the help of TCAD simulations, it is demonstrated that SEL is more sensitive with higher temperature. The conclusion is consistent with the previous works. The work is interesting and can provide information for radiation-harden design.

The main comments are listed as follows.

  1. The previous works have drawn that SEL is more sensitive to temperature. What’s the different work, which this paper has done?
  2. SEL is also sensitive to the distance between N-WELL and NMOS transistors. How the distance selected in the simulation work of this paper? It is appreciated to give more explanation about that.
  3. Line 28-29. The relationship between Celsius degree and absolute temperature maybe wrong. Please check it.
  4. Some writing errors need to be corrected. For example: line 47-49. Line 92-93. Line 153: Figure 3, is.
  5. Line 249. The current-time characteristic curve maybe not be got from Fig.10. The caption is same as that of Fig.9.

Reviewer 2 Report

Comments and Suggestions for Authors

Dear Authors, thank you for the interesting research. Please find below some comments-questions:

  • Pag. 2: please define hold voltage and current for readers not skilled in the matter;
  • Pag. 2, last row: which heavy-ion model have you used in TCAD software?
  • Pag. 3: please define SRH
  • Pag. 6, Figure 7: It is possibile to identify the various parasitic resistors sowhn in figure in the CMOS cross-section?
  • Pag. 8, first row below Table: not-clear English
  • Pag. 9, Conclusions: is there any experimental measurements (even yours or in literature) supporting the shown theoretical results?
